# Test/Retest Reliability and Validity of Remote vs. In-Person Anthropometric and Physical Performance Assessments in Cancer Survivors and Supportive Partners

**DOI:** 10.3390/cancers14041075

**Published:** 2022-02-21

**Authors:** Teri W. Hoenemeyer, William W. Cole, Robert A. Oster, Dorothy W. Pekmezi, Andrea Pye, Wendy Demark-Wahnefried

**Affiliations:** 1Department of Nutrition Sciences, University of Alabama at Birmingham (UAB), Birmingham, AL 35233, USA; colew14@uab.edu (W.W.C.); apye@uab.edu (A.P.); demark@uab.edu (W.D.-W.); 2O’Neal Comprehensive Cancer Center at UAB, Birmingham, AL 35233, USA; roster@uabmc.edu (R.A.O.); dpekmezi@uab.edu (D.W.P.); 3Department of Preventive Medicine, UAB School of Medicine, Birmingham, AL 35233, USA; 4Department of Health Behavior, UAB School of Public Health, Birmingham, AL 35233, USA

**Keywords:** cancer survivors, lifestyle, anthropometry, physical performance, remote assessment, virtual assessment, obesity

## Abstract

**Simple Summary:**

To expand the reach of lifestyle interventions among cancer survivors, in-person anthropometric and physical performance assessments were adapted to remote means and evaluated for feasibility, safety, validity, and reliability. Cancer survivors and supportive partners (*n* = 112) were approached to participate in three sessions (two remote and one in-person) of anthropometric and physical performance testing and results were compared. There was 98% uptake and no adverse events. ICCs for remote assessments ranged from moderate (8’ timed walk = 0.47), to strong (8’ get-up-and-go = 0.74), to very strong (30 s chair stand = 0.80; sit-and-reach = 0.86; 2 min step test = 0.87; back scratch = 0.90; weight = 0.93; and waist circumference = 0.98) (*p*-values < 0.001). One-hundred percent concordance was found for side-by-side and semi-tandem balance and 87.5–90.3% for tandem stances. No significant differences between remote and in-person assessments were found for weight, 8’ timed walk, and 8’ get-up-and-go. Remote anthropometric and physical performance assessments are reliable, valid, acceptable, and safe among cancer survivors and supportive partners.

**Abstract:**

(1) Background: Anthropometric and physical performance testing is commonly done in lifestyle research and is traditionally performed in-person. To expand the scalability of lifestyle interventions among cancer survivors, in-person assessments were adapted to remote means and evaluated for feasibility, safety, validity, and reliability. (2) Methods: Cancer survivors and supportive partners were approached to participate in three anthropometric and physical performance testing sessions (two remote/one in-person). Correlations, concordance, and differences between testing modes were evaluated. (3) Results: 110-of-112 individuals approached for testing participated (98% uptake); the sample was 78% female, 64% non-Hispanic White, of mean age 58 years and body mass index = 32.4 kg/m^2^. ICCs for remote assessments ranged from moderate (8’ walk = 0.47), to strong (8’ get-up-and-go = 0.74), to very strong (30 s chair stand = 0.80; sit-and-reach = 0.86; 2 min step test = 0.87; back scratch = 0.90; weight = 0.93; waist circumference = 0.98) (*p*-values < 0.001). Perfect concordance (100%) was found for side-by-side and semi-tandem balance, and 87.5–90.3% for tandem balance. No significant differences between remote and in-person assessments were found for weight, 8’ walk, and 8’ get-up-and-go. No adverse events occurred and 75% indicated no preference or preferred virtual testing to in-person. (4) Conclusions: Remote anthropometric and physical performance assessments are reliable, valid, acceptable, and safe among cancer survivors and supportive partners.

## 1. Introduction

Cancer survivors are at increased risk for second malignancies, heart disease, and functional decline—conditions for which a healthy diet, regular physical activity, and the avoidance of obesity have been proposed [1]. Hence, several lifestyle interventions have been evaluated in this vulnerable population over the past few decades [2,3]. Given that there are valid and reliable means to objectively assess physical performance, as well as body weight status, these outcomes are commonly included in lifestyle research focused on cancer survivors.

Anthropometric measurements and physical performance tests are key measures that objectively assess the risk of physical decline associated with unhealthy weight gain and the loss of strength, flexibility, and balance. Most often used in geriatric or rehabilitative care and traditionally administered face-to-face by trained staff, the Senior Fitness Test (SFT) battery has demonstrated validity and reliability in assessing overall functional fitness, as well as function in key domains, e.g., cardiorespiratory endurance, muscular endurance and strength, flexibility, balance, and motor coordination [4,5]. To date, however, the reliability and validity of administering the SFT in a remote, “virtual” setting remains relatively untested or partially tested (i.e., chair stand, get-up-and-go, 8’ walk) [6,7,8].

The clinical trials research community was profoundly challenged by the unexpected disruption brought about by the COVID-19 pandemic [9,10]. For many, clinical trials had to be delayed or terminated in deference to the need for participant safety—particularly in trials that enroll cancer survivors and others with immuno-compromised conditions who are disproportionately at higher risk of COVID-19 related complications [11,12,13,14]. Innovative, reliable, valid, and safe assessment methods became critically important for clinical trials to continue [15,16].

Remote video conferencing has long been used to deliver and assess educational programs, and its use for medical consultation began to emerge roughly a decade ago [17,18,19,20]. Therefore, when faced with the challenges of limiting COVID-19 exposure risk for participants and staff, clinical enterprises adapted rapidly to remote means, with anticipation that virtual care may be “here-to-stay”, given its convenience. Remote observation and assessment also hold great potential within the context of clinical trials for evaluating outcomes efficiently, reliably, and safely [21,22]. Moreover, remote assessments can greatly increase the reach of clinical trials and overcome two of the major barriers to participation: time and travel [23,24]. However, adaptation is still in its infancy.

The purpose of this sub-study was to: (1) assess the feasibility of remotely delivered “virtual” assessments, and (2) ascertain test/retest reliability and validity of remotely administered virtual anthropometric (weight, waist circumference) and physical performance tests (balance, back scratch, sit-and-reach, 30-s chair stand, 8’ timed walk and get-up and go, and 2 min step test) when compared to face-to-face, in-person testing. The hypothesis was that remotely delivered virtual assessments are feasible (i.e., uptake and completion rates of >80%, and safe as demonstrated by a lack of serious adverse events), reliable (i.e., demonstrating at least moderate levels of association and concordance between testing and retesting), and valid (i.e., demonstrating at least moderate levels of association and concordance between remote and in-person testing).

## 2. Materials and Methods

### 2.1. Study Design

This study sought to determine the feasibility, test/retest reliability and validity of remote methods for assessing anthropometric measures and physical performance testing in comparison to in-person testing. To do this, an ancillary study was conducted among participants enrolled in the Daughters, Dudes, Mothers, and Others (DUET) trial. DUET is a 6 month, single-blinded, 2 arm, randomized controlled trial (RCT) that evaluates a web-based lifestyle intervention against a wait-list control among 56 dyads (*n* = 112). Each dyad is comprised of a survivor of an obesity-related, early-stage cancer and their supportive partner, both of whom have obesity or are overweight, are insufficiently active, and consume suboptimal diets. The DUET RCT was conducted according to the guidelines of the Declaration of Helsinki and approved by the Institutional Review Board of The University of Alabama at Birmingham (UAB) (protocol code 300003882/Approval date: 28 October 2019) and is registered with ClinicalTrials.gov (NCT04132219). The DUET study design, methods and participants are described more fully by Pekmezi, et al. [25].

The baseline assessment of each anthropometric and physical performance measure was expanded to three consecutive assessments. These assessments were scheduled concurrently with both dyad members at approximately the same time of day and within a week (on average within the span of 3 consecutive days). First, two virtual assessment visits were observed remotely by trained assessors utilizing Zoom^®^ video conferencing, then, a third face-to-face, in-person home visit was scheduled. To keep the environment constant for this assessment, the home of one of the dyad members was used as the setting for all three assessments.

### 2.2. Participants

All 56 dyads (*n* = 112) enrolled in the DUET RCT were approached to participate in this ancillary study. The DUET RCT enrolled survivors of early stage endometrial, colorectal, breast, prostate, kidney, or ovarian cancer or multiple myeloma across Alabama, Mississippi, Tennessee, and North Carolina. Moreover, since this was a dyad-based intervention, participants needed a partner (with or without a cancer diagnosis) who was willing to participate in the trial and resided within 15 min driving distance from the survivor’s home. Eligibility criteria for dyads (survivor and partner) included reporting <2.5 cups per day of fruits and vegetables, <150 min a week of moderate-to-vigorous physical activity, no major physical limitations, and a Body Mass Index (BMI) >25 kg/m^2^. For this sub-study, dyads were also required to have access to internet-connected devices with a webcam and microphone, a scale to measure weight, a standard height (18 inch) non-upholstered chair, and a 12’ space to perform the physical performance tests.

### 2.3. Training

To ensure quality and consistency of data collection, three assessors completed 5 h supervised training sessions on collecting anthropometric and physical performance measurements utilizing Zoom^®^. A certification process followed, requiring that assessors conduct three simulated recorded assessments via Zoom^®^ on volunteers. The recordings were reviewed, graded (passing grade => 80%), and certified by two doctoral-level research scientists with expertise in diet and exercise trials. Assessors were evaluated on rapport with participants, use of health literate appropriate instructions, consistency of directions provided, and accuracy of measures documented.

### 2.4. Procedures

Two weeks prior to the first virtual assessment, the dyad received an email confirmation of the scheduled assessment and links to training videos on standardized anthropometric and physical performance assessments (https://youtu.be/G8p6g_VDzhw, accessed on 22 January 2022). Additionally, dyads were sent a package of remote assessment materials (8’ length of cord, two stickers, vinyl tape measure, two 4-1“x55” ribbons, felt tip marker, and a pre-paid return envelope) via US priority mail and a step-by-step instruction booklet of procedures. Two days prior to the first scheduled assessment, dyad members received a reminder call to review procedures to participate in the Zoom^®^-based assessment. For this assessment, dyad members who were not living in the same household but needed to be present for the assessments were reminded of the Center for Disease Control’s (CDC) COVID-19 precautions and UAB’s requirements for masking and social distancing [26].

On the scheduled dates for virtual assessments, assessors placed a password-protected Zoom^®^ call to the dyad, reviewed the instructions for the assessment measures with the pair and answered questions. To control for internet speed variance between sight and sound, virtual assessments were recorded (with participants’ permission), so that study staff could review the sessions later and assess time-based measures. The script followed by assessors during virtual assessments can be found in Appendix A and procedures followed before, during and after assessments are illustrated in Figure 1.

As noted previously, all Zoom^®^ sessions were recorded for review to increase accuracy of timed performance testing. Once assessors reviewed the videos, timed the tests, logged the data, and completed quality assurance tests, the recordings were deleted. Occurring in tandem, virtual assessments were scheduled during times when both dyad members could participate. This allowed one dyad member to assist, serve as a safety monitor, and hold the camera accurately while the other performed the tests while being instructed, observed, and recorded by the assessor via Zoom^®^. The data collected from all assessments were stored in REDCap (ver. 10.9.1) databases and were not accessed prior to assessments.

### 2.5. Anthropometric Measures

Standardized measures of weight and waist circumference [27] were collected and modified to allow for remote collection (Table 1). As noted in Table 1, scales were “zeroed out” prior to the participant stepping on them for the remote assessments. In addition, the same scale was used for in-person visits to assure consistency in weight measures across all assessments.

### 2.6. Physical Performance Measures

The Senior Fitness Test (SFT) battery objectively assesses physical performance in several domains, is sensitive to change, devoid of ceiling effects, and has normative scores [4,5,28,29]. Testing, typically done in-person, can include the following assessments: grip strength, arm curls, balance testing, sit-and-reach (flexibility), back scratch (flexibility), 30 s chair stand (lower body strength), 8’ get-up-and-go timed test (agility, dynamic balance), 8’ timed walk test (gait speed), and 2 min step test (endurance). For the DUET RCT, these tests were modified to allow for remote, “virtual” assessment (Table 2). Unfortunately, some elements of the SFT battery could not be readily adapted for remote assessment given budgetary constraints. For example, postage required to mail 5 and 8 lb. weights for the arm curl test were prohibitively expensive and, therefore, were omitted from remote assessments. Likewise, dynamometers to assess grip strength also exceeded the budget.

Along with SFT physical performance measures, balance was measured using the CDC advocated protocol [30] for side-by-side, semi-tandem, and tandem stances during which the camera was held by the partner, placed on the floor angled up, or placed on a table or chair across from the participant so that the full body, feet-to-shoulders, could be viewed. The test was conducted near a wall so that participants could steady themselves as needed. In addition, participants were instructed to keep one hand on the wall until the test began so the assessor could remotely observe when the timed test started.

Upon completion of the DUET study, we conducted an online debriefing survey among cancer survivors and their supportive partners. Of the 16 items, the survey posed five questions regarding the ease of viewing and the helpfulness of remote assessment instructional videos, the ease of connecting to Zoom^®^ and ease of completing the remote assessment. Five-point Likert scales were used with the following anchors: very easy (very helpful); easy (helpful); neutral; not easy (not helpful); not easy at all (not helpful at all), and the overall preference for in-person or remote assessments (anchors: no preference; in-person; virtual).

### 2.7. Data Analysis

The reliability and validity of the continuous anthropometric and physical performance measurements and scores were assessed using the Bland-Altman approach [31,32]. Reliability was evaluated between the two virtual assessments using intra-class correlation coefficient (ICC) analysis. This procedure also was employed to characterize associations between the virtual assessments and the standard in-person assessment to evaluate validity; specifically, the averaged mean of the two virtual assessments was compared to the value of the in-person assessment. Paired t-tests also were performed to test the mean differences between these assessments. Agreement (precision) of the individual measurements or scores was also assessed using the Bland-Altman 95% limits of agreement. The concordance of categorical balance scores was examined using McNemar’s test [33], first for Virtual Assessments 1 and 2, then for Virtual Assessment 1 to the in-person assessment and finally for Virtual Assessment 2 to the in-person assessment. Inter-rater reliability could not be examined, since one of the assessors performed most of the evaluations. Statistical tests were two-sided. Statistical significance was set at *p* < 0.05, with no adjustment for multiple testing. Reliability and validity analyses were performed first for the entire sample, then for cancer survivors, and, finally, for their partners. Analyses were performed using SAS, version 9.4 (SAS Institute, Inc.; Cary, NC, USA). In addition, data captured from the on-line debriefing survey were analyzed using descriptive statistics (frequencies and percentages).

### 2.8. Sample Size Considerations

The primary power calculation for the DUET RCT appears elsewhere [25]. While there are no generally agreed upon criteria for the required sample size for reliability and validity studies, a sample size of at least 50 participants has been suggested so that the Bland-Altman 95% limits of agreement can be precisely estimated [34,35].

## 3. Results

### 3.1. Demographics and Characteristics

Of the 112 participants (56 dyads) enrolled in the DUET trial who were approached to participate in this ancillary study to assess the reliability and validity of the virtual assessments as compared to in-person testing, all but two (one dyad) participated (*n* = 110) in at least one remote assessment. The overall sample of both cancer survivors and their partners was primarily urban dwelling (92%), female (78%), non-Hispanic White (NHW) (64%) individuals who had a mean age of 58 years and a BMI of 32.4 kg/m^2^. Moreover, roughly half were college graduates (53%) and were currently employed (55%), and of those who reported income, 72% indicated annual incomes of $50,000 or more.

Similar to the overall sample, survivors were found to be mostly urban dwelling (93%), female (87%), and NHW (64%), with an average age of 60 years, an average BMI of 31.8 kg/m^2^ and most often diagnosed with primary breast cancer (81%). Partners often cohabitated with the survivor (41%), and for those who did not, the overall average distance between dyad members was 7.1 miles. Partners were also primarily urban dwelling (91%), female (69%), and NHW (63%), with an average age of 56.5 years, and an average BMI of 33.0 kg/m^2^. Partners’ educational, employment, and income status were similar to survivors, as well as the overall sample. Of note, 13% (*n* = 7) of partners also reported a prior cancer diagnosis.

### 3.2. Validity and Reliability Results—Overall Sample

Of the 110 participants in this sample, 74 completed both remote assessments, as well as the in-person assessment; these data (less missing measurements) were used for the comparative analyses. Table 3 provides ICCs (along with 95% confidence intervals (CI)) with asterisks indicating the level of significance as indicators of strength of agreement between the two virtual assessments for the overall (all participants) sample. Of note, all ICCs were highly significant (*p* < 0.001), with correlations ranging from moderate (8’ walk, 0.47), to strong (8’ get-up-and-go, 0.74), to very strong (30 s chair stand, 0.80; sit-and-reach, 0.86; 2 min step test, 0.87; back scratch, 0.90; weight, 0.93; and waist circumference, 0.98).

Table 4 provides data supporting validity by showing ICCs (along with 95% CI’s) for comparisons between the means of the two virtual assessments and in-person assessment. Like the ICCs for reliability, these ICCs ranged from moderately strong (8’ walk, 0.65) to very strong (8’ get-up-and-go, 0.80; 2 min step test, 0.84; 30 s chair stand, 0.86; sit-and-reach, 0.89; back scratch, 0.95; waist circumference, 0.96; and weight, 0.98).

In addition, Table 4 shows the results of outcomes related to mean differences and *p*-values related to potential differences. Here, significant differences are not the desired outcome, and, indeed, we were unable to find significant differences for weight between the two virtual assessments, nor between the virtual assessments and the in-person assessment. Also, there was no significant difference between the two virtual assessments for waist circumference; however, a significant difference was detected between the virtual assessments and the in-person assessment. There were no significant differences found between the two virtual assessments nor between the virtual assessments and the in-person assessment for the 8’ walk. Additionally, there was no significant difference for the back scratch test between the two virtual assessments; however, a significant difference was detected between the virtual assessments and the in-person assessment. Conversely, for the 8’ get-up-and-go test, there was no significant differences found between the virtual assessments and the in-person assessment, but there was a significant difference found between the two virtual assessments. Significant differences between the two virtual assessments, as well as the virtual and in-person assessments, were found for the 30 s chair, the sit-and-reach, and the 2 min step test.

Finally, Figure 2 illustrates the results of the Bland-Altman testing and limits of agreement. The plots depict excellent agreement between virtual and in-person assessments with only a small proportion of observations falling outside of the limits of agreement for body weight (2.8%), agility (8.1%), strength (5.4%), and flexibility (5.5%).

### 3.3. Validity and Reliability Results—Survivors and Partners

Analyzed separately for survivors and partners, data for the anthropometric measures were corroborated and these tables are included as Appendix A. Similarly, in separate groups, data for the 8’ walk parallel the overall sample between the two virtual assessments and between the virtual assessments and the in-person, with no significant difference noted in either group. Between groups, there were fewer differences noted overall. Comparing the two virtual assessments, the only significant differences noted were for the 30 s chair stand and the 2 min step test.

Between the two virtual assessments, as well as for the virtual assessment compared to the in-person, the 30 s chair stand remained significantly different for each subgroup. While there was no significant difference found in the back scratch for the virtual assessment comparison, a significant difference existed for the survivor group when comparing the virtual assessments to the in-person assessment.

### 3.4. Validity and Reliability Results—Balance

With balance testing, no significant differences were found between the two virtual assessments, nor for the virtual and in-person assessments for any of the stances, i.e., side-by-side, semi-tandem or tandem. Additionally, categorical scoring (i.e., ability to hold the 10 s stance vs. not) was 100% concordant between the two virtual assessments, as well as across the virtual assessments and the in-person assessment for both the side-by-side and semi-tandem tests. For tandem testing in all participants, scores were 86.1% concordant between the two virtual assessments, and ranged from 87.5 to 90.3% concordance between each virtual test and the in-person assessment. In the survivor group, concordance between the two virtual assessment scores was 86.1%, whereas concordance ranged from 88.9% to 91.7% between each virtual test and the in-person assessment, whereas for partners, these values for concordance were 86.2% and 83.4–91.7%, respectively.

### 3.5. Feasibility Benchmarks

In conducting this study, we found that all participants had the resources needed to participate in the remote assessments, including access to the Internet, smart phones or laptop computers with cameras, and the ability to use the Zoom™ app, and that uptake was exceedingly high, i.e., 98% (*n* = 110). In fact, only one cancer survivor–partner dyad did not participate in this sub-study, and was unable to due to scheduling conflicts. Furthermore, of the 220 possible virtual assessments, 204 (92.7%) were completed (with incomplete assessments largely due to poor internet connectivity). Finally, the average time for completing a virtual assessment was 25 min when compared to a mean of 35 min for the in-person assessment (not inclusive of travel time).

Debriefing survey results indicated that 77% of respondents reported that instructional videos for remote assessments were “easy to very easy” to view and 85% conveyed that they were “helpful to very helpful”. Moreover, Zoom^®^ assessments were viewed as “easy to very easy” by 91% of respondents and 83% indicated that completing the remote assessment was “easy to very easy”. Sixty percent of respondents voiced “no preference” for assessment mode, whereas 24% preferred in-person and 16% preferred virtual assessments. Importantly, there were no adverse events reported during these assessments.

## 4. Discussion

This is the first study to perform both anthropometric and physical performance testing using remote means in cancer survivors. As such, it represents a significant development for both observational and interventional studies that assess lifestyle factors, as well as associated health outcomes in this important patient population. Indeed, as lifestyle interventions among cancer survivors have developed over the years, there has been an ever-increasing trend to pursue home-based interventions [36,37], especially since time and travel are noted barriers to participation [23,24]. Tailored print and telephone-delivered interventions were employed and paved the way to web-based interventions that are scalable and able to reach cancer survivors living in close proximity to the clinic and those living more remotely; however, the means to assess these interventions have not kept step.

Blair et al. [8] were the first to report the evaluation of remote means of physical performance testing in a carefully phased approach and exclusively among older cancer survivors using the 30 s chair and the 8’ get-up-and-go. Phase I of their proof of concept was performed among 10 cancer survivors of mean age 70.5 years and showed promise; however, the phased investigation was halted due to the COVID-19 pandemic. While the pandemic negatively affected the progress of Blair et al., it provided impetus for others. In a published abstract, Winters-Stone et al. [7] reported preliminary findings from a study that focused on inter- and intra-rater reliability and validity of a limited set of physical performance tests that were adapted for remote assessment, as compared to in-person tests among older cancer survivors. Pearson-product moment correlations suggested excellent intra-rater reliability for the timed chair stand (0.88), the 4 m walk (0.90), and the 8’ get-up-and-go (0.96), whereas inter-rater reliability for these tests was lower, i.e., 0.14, 0.40, and 0.99, respectively. Correlation coefficients between virtual and in-person tests were 0.81 for the timed chair stand and 0.78 for the 4 m walk [7]. Given these strong correlations which support reliability for remote testing (especially if performed by the same personnel), Winters-Stone and colleagues have gone forward and used remote assessments in home-based strength training interventions among both breast and prostate cancer survivors, crediting these developments to ultimately improving retention [38]. While the physical performance tests that were studied by Winters-Stone are slightly different from the tests evaluated in the current study (i.e., the 4 m walk versus the 8’ walk, and the timed chair stand vs. the 30 s chair stand), there is relative agreement on the statistics related to reliability. Moreover, there are similarities in the strong correlations found between virtual and in-person chair stand and walk tests in the study performed by Winters-Stone et al. and the ICCs emanating from the current study—both of which support validity. However, in applying a Bland-Altman analysis (an additional step performed in the current study), significant differences were found between virtual and in-person chair stands. While this threat to validity may be explained by the omission of statistical analysis in the work of Winters-Stone et al., it is more likely that, even if such testing were performed, the former study had a longer lag time between assessments (2 weeks vs. 3.3 days) and included one fewer assessment. Indeed, the significant differences revealed by Bland-Altman analyses between virtual and in-person testing in the current study for the 30 s stand, 2 min step test, and flexibility testing could largely be explained by practice effects resulting from the rapid test–retest nature of the protocol, which was focused largely on the primary outcome of DUET, i.e., weight [39].

In the current study, exceptionally strong ICCs ranging from 0.93 to 0.98 were found between virtually assessed body weights and between remote vs. in-person measures. Moreover, no significant differences existed across measures suggesting that the methods that we employed were both valid and reliable. Given a substantial literature on the inaccuracy and underreporting of self-reported weight [40,41,42,43], the procedure outlined in this study, could be helpful if utilized in other studies—especially those in which budget constraints preclude the use of Bluetooth or Wi-Fi enabled scales. Waist circumference, another measure for which there is substantial evidence for under-reporting [44], also could benefit from measurement under the watchful eye of videoconferencing. A recent study conducted on a sample of 15 adult–adult dyads and 10 parent–child dyads found no clinically meaningful differences between waist circumference measures collected via virtual assessment when compared to in-person assessment [45]. These findings, along with ours, are highly suggestive that observation of waist circumference measurements via camera may be a way to reduce systemic self-reported bias and achieve reliable results that are highly correlated with assessor-based measures. Exciting new developments in assessing body size with three-dimensional software allowing measurements being uploaded to a smartphone are also on the research horizon [46].

Additionally, the current study fills gaps in the research literature by providing more extensive physical performance tests that not only measure agility and lower body strength, but which also measure balance, flexibility (back scratch and sit-and-reach), and fitness or endurance (2 min step test). In these areas there is a dearth of studies among cancer survivors, and only one reported study of remotely assessed 2 min step tests in a non-cancer population (*n* = 55 older veterans), with ICCs of 0.98 in the same remote assessments evaluated by two assessors [6]. In contrast, the strong ICCs found in the current study between the two virtual assessments (i.e., 0.87) and the virtual and in-person assessments (0.84) in the current study are perhaps a stronger test of reliability, since measures were conducted on different days. These results also offer information on validity, since test results were compared to in-person assessments.

In terms of feasibility, there was exceptionally high uptake and completion of this sub-study. In general, participants were willing and able to complete the assessments virtually, with <8% of assessments needing to be rescheduled due to internet connectively problems. In comparison, the study by Ogawa et al. [6] among older veterans found that technical difficulties interfered with 5% of the virtual physical performance assessments (3 out of 60). Home environment challenges (e.g., poor lighting, poor camera angle) were cited during those virtual assessments [6] but not in the current study, perhaps due to the provision of related instructional videos and written materials, for which the majority of participants rated as helpful and easy to access. Moreover, 83% of participants in the current study reported that the remote assessments were easy to complete. Further, while the vast majority (60%) of participants claimed no preference in the mode of assessment, those claiming a preference for in-person versus remote were roughly comparable: 24 vs. 16%. Finally, our most important feasibility finding was that no adverse events or injuries were reported during the test period. We attribute this safety to having a partner present for spotting and the conduct of testing by walls that could be used to provide support, as well as assessors who were attuned to participant status, such as ensuring that tests were performed in proper shoes on non-slip flooring, and testing area was cleared of obstacles (including pets).

### Study Limitations

While the abbreviated assessment timeline for the current study likely improved findings on reliability and validity for some measures (such as weight), it may also have interjected bias for others. Practice effect, or the systematic improvement in performance that occurs with repeated testing, has been found in past studies with the four-square step test, timed up and go, and 10 m walk test [47]. Future evaluations may consider randomizing or counterbalancing the order of remote and in-person assessments to alleviate this potential source of bias. As for other limitations, inter-/intra-rater variance could not be examined because most of the assessments were done by one assessor due to staffing shortages. Additionally, our study sample, while representative of the general population in terms of race and ethnicity, included a higher proportion of white females (and among cancer survivors, most had a previous diagnosis of breast cancer). Perhaps even more importantly, proportionately, the sample had higher incomes and was more highly educated. Thus, it is unknown how findings can be generalized to racial/ethnic minorities, males, those of lower socio-economic status and survivors of other cancers.

It also should be noted that our study did not attempt measures of arm and grip strength because of economic constraints; thus, this is an area in which future studies could contribute by devising creative strategies to meet this need. For example, Ogawa et al. [6] used filled milk jugs to remotely conduct arm curl tests among older veterans—a method that could be employed among cancer survivors. Moreover, less expensive dynamometers are now being marketed and should be evaluated for research use.

Finally, while this sub-study explored measures that potentially could translate to an oncology clinical practice (e.g., measures of weight and waist circumference), the findings may have more relevance to the research setting. Therefore, additional adaptations and research therefore may be necessary for translation.

## 5. Conclusions

The development of remote assessment approaches for use in lifestyle research among cancer survivors is important, not only as a means to continue research efforts during the pandemic, but also as a means to improve the overall accrual and intervention scalability. Allowing individuals to participate in clinical trials from the comfort of their own homes reduces the time and travel concerns that serve as substantial barriers to recruitment, particularly for medically underserved populations that suffer disproportionate burden, and represent the very populations that clinical trials aim to serve. This research identified new methods for remotely assessing anthropometric and physical performance measures that are acceptable, safe, and reliable. Further research is needed to either corroborate or refute these results, as well as to develop feasible, reliable, and valid remote protocols for measuring other physical performance domains, such as upper arm and grip strength.

## Figures and Tables

**Figure 1 cancers-14-01075-f001:**
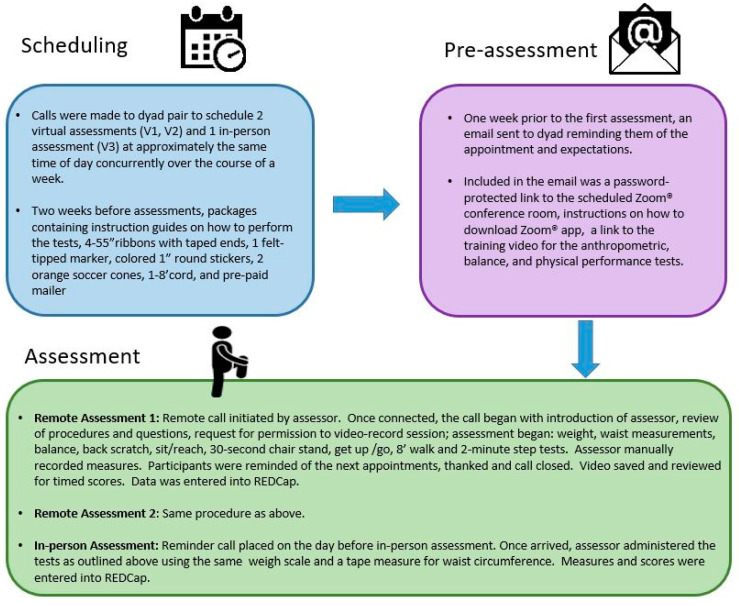
Overview of remote and in-person assessment procedures: illustrates steps taken before, during and after remote and in-person assessments.

**Figure 2 cancers-14-01075-f002:**
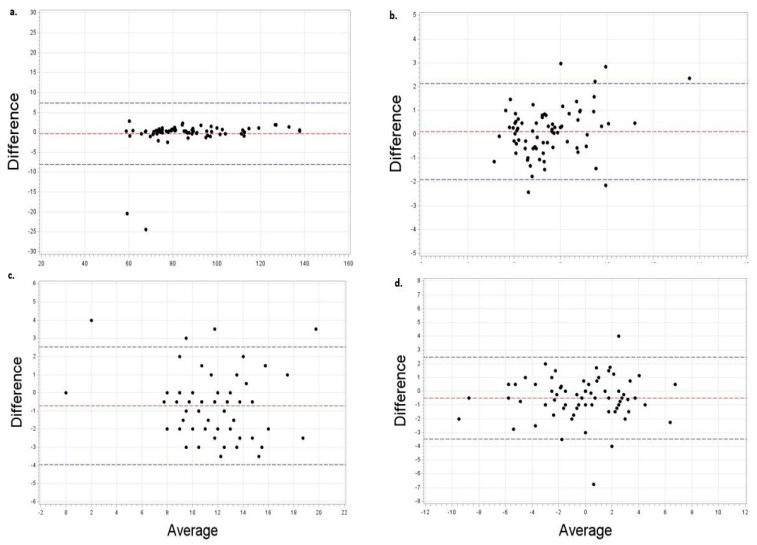
Bland Altman plots depicting agreement between virtual and in-person assessments: illustrates (**a**) body weight assessments with 2.1% falling outside limits of agreement; (**b**) agility (Get-up-and-go) with 8.1% falling outside limits of agreement; (**c**) strength (30-s chair stand) with 5.4% falling outside limits of agreement and (**d**) flexibility (Sit-and-reach) with 5.5% falling outside limits of agreement.

**Table 1 cancers-14-01075-t001:** Anthropometric measures and post-COVID-19 modifications.

Pre-COVID-19 Measure	COVID-19 Modification
Weight assessed in light clothing, without shoes, on a calibrated scale set on a hard surface observed in-person by assessor. Weight is measured twice, averaged, and recorded.	Modification for virtual assessments 1 and 2: participant’s scale is used; camera is positioned so assessor can see scale face. Participant is asked to “zero out” the scale prior to stepping on it. Modification for in-person assessment: assessor weighs the participants on the same scale as used in V1, V2.
Waist circumference assessed by assessor using a non-stretch tape measure, at level of umbilicus and upon exhale.	Modification for virtual assessments 1 and 2: Camera is place on floor angled up to capture midsection of participant and partner or placed on table/chair facing both so mid-section is in frame. Assessor instructs partner to place the taped end of the ribbon on the umbilicus; holding taped end in place, partner wraps the ribbon snugly around the waist (shirt lifted) placing ribbon parallel to the ground. Participant is asked to rotate in front of the camera to assure the ribbon is flat against the skin and parallel to the ground. Once proper technique is confirmed by the assessor, participant is asked to inhale, and, upon exhale, partner marks the point where the ribbon overlaps. The test is repeated and both ribbons are mailed back to assessor to be measured.

**Table 2 cancers-14-01075-t002:** Physical performance measures [28,29] and post-COVID-19 modifications: virtual assessments 1 and 2 conducted remotely via Zoom^®^ video conference; 3rd assessment conducted in-person.

Original Measure (All of Which Have Proven Reliability and Validity)	Adaptations of the Reference Test to Permit Virtual Assessments
Sit-and-reach: a test of flexibility [28].	The camera is placed on floor at a 90-degree angle with direct view of seated participant with bent and extended legs in frame; assessor guides participant and partner in measuring distance and participant through procedure.
30 s chair stand: a test of lower body strength [28].	The camera is placed on floor angled to see participant in both seated and standing positions or held by partner; assessor instructs participant to place the chair against a wall to prevent it from moving during test and guides the participant and partner through test procedure.
Back scratch: a test of lower body strength [28].	The camera is placed on table or counter, angled to see participant’s back and observe the partner’s measurements, as well as which side is being measured, using standard vinyl tape measure included in study supplies package. Assessor guides participant and partner on test protocol.
8’ Get-up-and-go (TUG): a test of lower body strength [29]	The camera is placed on floor to observe partner measure 8’ distance using 8’ cord included in study materials (along with color stickers and orange cones to serve as “markers”). Camera is positioned on floor or held by partner so that seated participant and orange cone marking 8’ end point are in frame. Assessor gives signal to “go” after ensuring participant is correct starting position. Timed score is recorded after viewing time stamp on video recording.
Timed 8’ Walk: a test to assess gait [28].	Assessor instructs partner to move cone out horizontally from previous test and place the other cone adjacent for mark “finish line’ with space wide enough for passage. The camera is placed on floor at a distance that captures the chair and the orange cones that signify the “finish line”. Time is stopped as soon as participant breaks the plane of the “finish line”. Timed score is recorded after viewing time stamp on video recording.
2 min step test: a test to assess endurance [28].	The camera is placed on floor angled towards wall to ensure proper measurement and sticker placement, i.e., a sticker is placed on the wall that corresponds to the measured midpoint between the trochanter and patella, and then the distance between that midpoint to the floor below (see script). The camera is then angled so that a full view of participant (head-to-toe) is in frame. Assessor instructs participant not to talk during test, to draw-up knee to the height of the sticker with each step, and to rest as needed. Partner is instructed to spot as needed, Score on completed steps is recorded.

**Table 3 cancers-14-01075-t003:** Comparison of anthropometric and physical performance data obtained between two sequential virtual assessments (V1, V2) (reliability) for all DUET participants.

Measure	Virtual 1 (V1) Mean (SD)	Virtual 2 (V2) Mean (SD)	ICC (95% CI) (V1 vs. V2) *	Mean Difference (V1–V2)	Limits of Agreement (V1–V2)	*p*-Value
Weight (kg) [*n* = 71]	86.2 (21.0)	87.4 (18.8)	0.93 (0.89, 0.96)	−1.2 (7.5)	−15.9, 13.5	0.18
Waist Circumference (cm) [*n* = 73]	107.9 (14.9)	107.4 (15.1)	0.98 (0.97, 0.99)	0.5 (2.8)	−4.9, 6.0	0.11
30 s Chair stand (reps) [*n* = 74]	10.7 (3.0)	12.0 (3.5)	0.80 (0.69, 0.87)	−1.3 (2.1)	−5.4, 2.9	<0.001
8’ Get Up/Go (sec to 10th) [*n* = 74]	7.8 (2.1)	7.4 (1.6)	0.74 (0.62, 0.83)	0.5 (1.4)	−2.3, 3.2	0.006
8’ Walk (sec to 10th) [*n* = 74]	2.2 (0.5)	2.2 (0.4)	0.47 (0.27, 0.63)	0.0 (0.5)	−0.9, 1.0	0.64
Sit & Reach (cm) [*n* = 73]	−0.7 (3.5)	−0.3 (3.2)	0.86 (0.78, 0.91)	−0.4 (1.8)	−4.0, 3.1	0.047
Back scratch (cm) [*n* = 72]	−3.5 (3.4)	−3.4 (3.4)	0.90 (0.84, 0.93)	−0.0 (1.5)	−3.0, 3.0	0.96
2-min step test (# steps) [*n* = 74]	77.3 (23.3)	82.4 (22.8)	0.87 (0.80, 0.92)	−5.1 (11.8)	−28.1, 18.0	<0.001

* ICC = intra-class correlation coefficient, CI = confidence interval.

**Table 4 cancers-14-01075-t004:** Comparison of anthropometric and physical performance data obtained between the average of virtual assessments and in-person measures (validity) for all DUET participants.

Measure	(V1 + V2)/2 Mean (SD)	In-Person Mean (SD)	ICC and 95% CI (Average Virtual vs. In-Person) *	Mean Difference (Average Virtual vs. In-Person)	Limits of Agreement (Average Virtual vs. In-Person)	*p*-Value
Weight (kg) [*n* = 71]	86.8 (19.5)	87.2 (18.6)	0.98 (0.97, 0.99)	−0.4 (3.9)	−8.0, 7.3	0.43
Waist Circumference (cm) [*n* = 73]	107.7 (14.9)	103.6 (15.5)	0.96 (0.93, 0.97)	4.1 (4.6)	−4.9, 13.0	<0.001
30 s Chair stand (reps) [*n* = 74]	11.3 (3.1)	12.1 (3.3)	0.86 (0.79, 0.92)	−0.7 (1.7)	−4.0, 2.5	<0.001
8’ Get Up/Go (sec to 10th) [*n* = 74]	7.6 (1.7)	7.5 (1.4)	0.80 (0.70, 0.87)	0.1 (1.0)	−1.9, 2.1	0.32
8’ Walk (sec to 10th) [*n* = 74]	2.2 (0.4)	2.2(0.4)	0.65 (0.49, 0.76)	0.0 (0.3)	−0.6, 0.7	0.35
Sit & Reach (cm) [*n* = 73]	−0.5 (3.2)	−0.0 (3.2)	0.89 (0.82, 0.93)	−0.5 (1.5)	−3.5, 2.5	0.009
Back scratch (cm) [*n* = 72]	−3.5 (3.3)	−3.1 (3.2)	0.95 (0.92, 0.97)	−0.4 (1.0)	−2.4, 1.6	0.003
2-min step test (# steps) [*n* = 74]	79.8 (22.3)	85.3 (25.9)	0.84 (0.76, 0.90)	−5.5 (13.9)	−32.8, 21.9	0.001

* CI = confidence interval.

## Data Availability

The data presented in this study are available in this article and Appendix A.

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
