# Peer review of "Test/Retest Reliability and Validity of Remote vs. In-Person Anthropometric and Physical Performance Assessments in Cancer Survivors and Supportive Partners"

_cancers, 2022, doi:10.3390/cancers14041075_

Round 1

Reviewer 1 Report

This paper addresses an important and timely issue, that of finding ways to accurately assess health and physical performance variables in remote conditions when in-person assessment is not feasible.  The paper is very well-written and overall well-presented, covering the basic elements of scientific research -- appropriate and timely review of the literature, appropriate design and analysis, and clear and justifiable conclusions. 

Two questions/concerns that I'd like to see addressed are:

1) An explanation for the loss participant numbers from onset to data analysis.  Authors report that all but two of the original 112 participants enrolled in the study completed the assessments (See Results Section 3.1), yet the results shown in Tables 3 and 4 indicate N sizes ranging from 71-74.  I did not find an explanation for loss of participants during the study, which should be addressed if it wasn't (perhaps it was and I missed it).

2) There also could be a question about whether or not the 'in-person' assessors were aware of the remote test results (Virtual 1 and Virtual 2 tests) or if they were 'blind' to the previous scores.  If assessors were aware of previous scores, this should be included in the limitation discussion, since it could possbily result in unintentional scoring bias.  

Finally, I found the description of how to determine knee height in the 2-minute step test a little hard to follow.  If possible, suggest trying to find a clearer way of describing this procedure.  

Overall, great job in carrying out this important work.  

Author Response

Reviewer 1 An explanation for the loss participant numbers from onset to data analysis.  Authors report that all but two of the original 112 participants enrolled in the study completed the assessments (See Results Section 3.1), yet the results shown in Tables 3 and 4 indicate N sizes ranging from 71-74.  I did not find an explanation for loss of participants during the study, which should be addressed if it wasn't (perhaps it was and I missed it).

Response: Thank you for the opportunity to address and clarify this.  We approached 112 potential participants for this sub-study of whom 110 agreed to complete at least 1 virtual assessment.  Of the 110, 74 completed both virtual assessments and the in-person assessment and these data were used for our analyses excluding missing measurements.  We have clarified this in the Results section (Lines 228, 244).

There also could be a question about whether or not the 'in-person' assessors were aware of the remote test results (Virtual 1 and Virtual 2 tests) or if they were 'blind' to the previous scores.  If assessors were aware of previous scores, this should be included in the limitation discussion, since it could possibly result in unintentional scoring bias.

Response: Previous scores were not viewed prior to any assessments.. We have clarified this in the Methods section (Line 160).

Finally, I found the description of how to determine knee height in the 2-minute step test a little hard to follow.  If possible, suggest trying to find a clearer way of describing this procedure.

Response: We have clarified this procedure in the Supplemental script and in Table 2 of the Methods section.

Reviewer 1 An explanation for the loss participant numbers from onset to data analysis.  Authors report that all but two of the original 112 participants enrolled in the study completed the assessments (See Results Section 3.1), yet the results shown in Tables 3 and 4 indicate N sizes ranging from 71-74.  I did not find an explanation for loss of participants during the study, which should be addressed if it wasn't (perhaps it was and I missed it).

Response: Thank you for the opportunity to address and clarify this.  We approached 112 potential participants for this sub-study of whom 110 agreed to complete at least 1 virtual assessment.  Of the 110, 74 completed both virtual assessments and the in-person assessment and these data were used for our analyses excluding missing measurements.  We have clarified this in the Results section (Lines 228, 244).

There also could be a question about whether or not the 'in-person' assessors were aware of the remote test results (Virtual 1 and Virtual 2 tests) or if they were 'blind' to the previous scores.  If assessors were aware of previous scores, this should be included in the limitation discussion, since it could possibly result in unintentional scoring bias.

Response: Previous scores were not viewed prior to any assessments.. We have clarified this in the Methods section (Line 160).

Finally, I found the description of how to determine knee height in the 2-minute step test a little hard to follow.  If possible, suggest trying to find a clearer way of describing this procedure.

Response: We have clarified this procedure in the Supplemental script and in Table 2 of the Methods section.

Reviewer 1 An explanation for the loss participant numbers from onset to data analysis.  Authors report that all but two of the original 112 participants enrolled in the study completed the assessments (See Results Section 3.1), yet the results shown in Tables 3 and 4 indicate N sizes ranging from 71-74.  I did not find an explanation for loss of participants during the study, which should be addressed if it wasn't (perhaps it was and I missed it).

Response: Thank you for the opportunity to address and clarify this.  We approached 112 potential participants for this sub-study of whom 110 agreed to complete at least 1 virtual assessment.  Of the 110, 74 completed both virtual assessments and the in-person assessment and these data were used for our analyses excluding missing measurements.  We have clarified this in the Results section (Lines 228, 244).

There also could be a question about whether or not the 'in-person' assessors were aware of the remote test results (Virtual 1 and Virtual 2 tests) or if they were 'blind' to the previous scores.  If assessors were aware of previous scores, this should be included in the limitation discussion, since it could possibly result in unintentional scoring bias.

Response: Previous scores were not viewed prior to any assessments.. We have clarified this in the Methods section (Line 160).

Finally, I found the description of how to determine knee height in the 2-minute step test a little hard to follow.  If possible, suggest trying to find a clearer way of describing this procedure.

Response: We have clarified this procedure in the Supplemental script and in Table 2 of the Methods section.

Reviewer 1 An explanation for the loss participant numbers from onset to data analysis.  Authors report that all but two of the original 112 participants enrolled in the study completed the assessments (See Results Section 3.1), yet the results shown in Tables 3 and 4 indicate N sizes ranging from 71-74.  I did not find an explanation for loss of participants during the study, which should be addressed if it wasn't (perhaps it was and I missed it).

Response: Thank you for the opportunity to address and clarify this.  We approached 112 potential participants for this sub-study of whom 110 agreed to complete at least 1 virtual assessment.  Of the 110, 74 completed both virtual assessments and the in-person assessment and these data were used for our analyses excluding missing measurements.  We have clarified this in the Results section (Lines 228, 244).

There also could be a question about whether or not the 'in-person' assessors were aware of the remote test results (Virtual 1 and Virtual 2 tests) or if they were 'blind' to the previous scores.  If assessors were aware of previous scores, this should be included in the limitation discussion, since it could possibly result in unintentional scoring bias.

Response: Previous scores were not viewed prior to any assessments.. We have clarified this in the Methods section (Line 160).

Finally, I found the description of how to determine knee height in the 2-minute step test a little hard to follow.  If possible, suggest trying to find a clearer way of describing this procedure.

Response: We have clarified this procedure in the Supplemental script and in Table 2 of the Methods section.

Reviewer 1 An explanation for the loss participant numbers from onset to data analysis.  Authors report that all but two of the original 112 participants enrolled in the study completed the assessments (See Results Section 3.1), yet the results shown in Tables 3 and 4 indicate N sizes ranging from 71-74.  I did not find an explanation for loss of participants during the study, which should be addressed if it wasn't (perhaps it was and I missed it).

Response: Thank you for the opportunity to address and clarify this.  We approached 112 potential participants for this sub-study of whom 110 agreed to complete at least 1 virtual assessment.  Of the 110, 74 completed both virtual assessments and the in-person assessment and these data were used for our analyses excluding missing measurements.  We have clarified this in the Results section (Lines 228, 244).

There also could be a question about whether or not the 'in-person' assessors were aware of the remote test results (Virtual 1 and Virtual 2 tests) or if they were 'blind' to the previous scores.  If assessors were aware of previous scores, this should be included in the limitation discussion, since it could possibly result in unintentional scoring bias.

Response: Previous scores were not viewed prior to any assessments.. We have clarified this in the Methods section (Line 160).

Finally, I found the description of how to determine knee height in the 2-minute step test a little hard to follow.  If possible, suggest trying to find a clearer way of describing this procedure.

Response: We have clarified this procedure in the Supplemental script and in Table 2 of the Methods section.

Reviewer 1 An explanation for the loss participant numbers from onset to data analysis.  Authors report that all but two of the original 112 participants enrolled in the study completed the assessments (See Results Section 3.1), yet the results shown in Tables 3 and 4 indicate N sizes ranging from 71-74.  I did not find an explanation for loss of participants during the study, which should be addressed if it wasn't (perhaps it was and I missed it).

Response: Thank you for the opportunity to address and clarify this.  We approached 112 potential participants for this sub-study of whom 110 agreed to complete at least 1 virtual assessment.  Of the 110, 74 completed both virtual assessments and the in-person assessment and these data were used for our analyses excluding missing measurements.  We have clarified this in the Results section (Lines 228, 244).

There also could be a question about whether or not the 'in-person' assessors were aware of the remote test results (Virtual 1 and Virtual 2 tests) or if they were 'blind' to the previous scores.  If assessors were aware of previous scores, this should be included in the limitation discussion, since it could possibly result in unintentional scoring bias.

Response: Previous scores were not viewed prior to any assessments.. We have clarified this in the Methods section (Line 160).

Finally, I found the description of how to determine knee height in the 2-minute step test a little hard to follow.  If possible, suggest trying to find a clearer way of describing this procedure.

Response: We have clarified this procedure in the Supplemental script and in Table 2 of the Methods section.

Reviewer 1 An explanation for the loss participant numbers from onset to data analysis.  Authors report that all but two of the original 112 participants enrolled in the study completed the assessments (See Results Section 3.1), yet the results shown in Tables 3 and 4 indicate N sizes ranging from 71-74.  I did not find an explanation for loss of participants during the study, which should be addressed if it wasn't (perhaps it was and I missed it).

Response: Thank you for the opportunity to address and clarify this.  We approached 112 potential participants for this sub-study of whom 110 agreed to complete at least 1 virtual assessment.  Of the 110, 74 completed both virtual assessments and the in-person assessment and these data were used for our analyses excluding missing measurements.  We have clarified this in the Results section (Lines 228, 244).

There also could be a question about whether or not the 'in-person' assessors were aware of the remote test results (Virtual 1 and Virtual 2 tests) or if they were 'blind' to the previous scores.  If assessors were aware of previous scores, this should be included in the limitation discussion, since it could possibly result in unintentional scoring bias.

Response: Previous scores were not viewed prior to any assessments.. We have clarified this in the Methods section (Line 160).

Finally, I found the description of how to determine knee height in the 2-minute step test a little hard to follow.  If possible, suggest trying to find a clearer way of describing this procedure.

Response: We have clarified this procedure in the Supplemental script and in Table 2 of the Methods section.

Reviewer 1 An explanation for the loss participant numbers from onset to data analysis.  Authors report that all but two of the original 112 participants enrolled in the study completed the assessments (See Results Section 3.1), yet the results shown in Tables 3 and 4 indicate N sizes ranging from 71-74.  I did not find an explanation for loss of participants during the study, which should be addressed if it wasn't (perhaps it was and I missed it).

Response: Thank you for the opportunity to address and clarify this.  We approached 112 potential participants for this sub-study of whom 110 agreed to complete at least 1 virtual assessment.  Of the 110, 74 completed both virtual assessments and the in-person assessment and these data were used for our analyses excluding missing measurements.  We have clarified this in the Results section (Lines 228, 244).

There also could be a question about whether or not the 'in-person' assessors were aware of the remote test results (Virtual 1 and Virtual 2 tests) or if they were 'blind' to the previous scores.  If assessors were aware of previous scores, this should be included in the limitation discussion, since it could possibly result in unintentional scoring bias.

Response: Previous scores were not viewed prior to any assessments.. We have clarified this in the Methods section (Line 160).

Finally, I found the description of how to determine knee height in the 2-minute step test a little hard to follow.  If possible, suggest trying to find a clearer way of describing this procedure.

Response: We have clarified this procedure in the Supplemental script and in Table 2 of the Methods section.

Reviewer 1 An explanation for the loss participant numbers from onset to data analysis.  Authors report that all but two of the original 112 participants enrolled in the study completed the assessments (See Results Section 3.1), yet the results shown in Tables 3 and 4 indicate N sizes ranging from 71-74.  I did not find an explanation for loss of participants during the study, which should be addressed if it wasn't (perhaps it was and I missed it).

Response: Thank you for the opportunity to address and clarify this.  We approached 112 potential participants for this sub-study of whom 110 agreed to complete at least 1 virtual assessment.  Of the 110, 74 completed both virtual assessments and the in-person assessment and these data were used for our analyses excluding missing measurements.  We have clarified this in the Results section (Lines 228, 244).

There also could be a question about whether or not the 'in-person' assessors were aware of the remote test results (Virtual 1 and Virtual 2 tests) or if they were 'blind' to the previous scores.  If assessors were aware of previous scores, this should be included in the limitation discussion, since it could possibly result in unintentional scoring bias.

Response: Previous scores were not viewed prior to any assessments.. We have clarified this in the Methods section (Line 160).

Finally, I found the description of how to determine knee height in the 2-minute step test a little hard to follow.  If possible, suggest trying to find a clearer way of describing this procedure.

Response: We have clarified this procedure in the Supplemental script and in Table 2 of the Methods section.

Reviewer 2 Report

Thank you for the opportunity to review this paper. I congratulate the authors on completing a much needed and timely piece of work. This paper examines the test-retest reliability of multiple anthropometric and physical tests when conducted by virtual means. It will be an essential read to many in the field who are now grappling with remote delivery of cancer rehabilitation and therefore will be highly cited. The detail provided in the methods is excellent and will allow the testing procedures to be replicated exactly in future trials. The Bland and Altman plots presented add to the strength of the statistical analysis and the opportunity for the reader to understand the data. It is well written and pragmatic in its presentation. I have no required changes and thank the authors for this essential contribution to the literature. 

Author Response

Thank you for your comments.

Reviewer 3 Report

Thank you for the opportunity to read this  publication. I have to appreciate a novelity of this publication and a lot of work which was done. It is  well-statistically processed method of measuring anthropometric and physical values in the home environment, which is especially beneficial in the new Covid-19  era.  However there should be some improvement.  The description of the measurement method lacked an information how long the virtual and personal consultation itself takes. For the readers - clinical oncologists, the description of the method seems to be very long. There is completely missing contribution of this  method for clinical use for cancer survivors. The aim of this new method – in future – find the new way how to make proper excercise under the supervision of a rehabilitation worker to improve physical coordination of patients, or how to reduce weight, other?  

Author Response

The description of the measurement method lacked an information how long the virtual and personal consultation itself takes. For the readers - clinical oncologists, the description of the method seems to be very long. Response: Thank you for the opportunity to respond.  The average time to complete the two virtual assessments was 25 minutes and the in-person assessment, 35 minutes.  We have added this information to the Results section. (Line 317)

There is completely missing contribution of this method for clinical use for cancer survivors. The aim of this new method – in future – find the new way how to make proper exercise under the supervision of a rehabilitation worker to improve physical coordination of patients, or how to reduce weight, other?   Response:  Cancers is a journal for which the readership is comprised not only of clinicians and clinician-scientists, but also basic and translational investigators. This manuscript was submitted for consideration in a special issue on the “Role of Lifestyle-related Factors in Cancer Survivorship,” for the purpose of disseminating our findings to cancer researchers who conduct anthropometric and physical function assessments in order to evaluate these outcomes within the context of clinical trials of lifestyle interventions.  While our hope is that oncologists find components of this study translatable to their practices (e.g., weight/waist measurements collected during telemedicine visits), our findings are perhaps more applicable in a research environment.  This approach is clarified in the Discussion section of the manuscript. (Line 443).

The description of the measurement method lacked an information how long the virtual and personal consultation itself takes. For the readers - clinical oncologists, the description of the method seems to be very long. Response: Thank you for the opportunity to respond.  The average time to complete the two virtual assessments was 25 minutes and the in-person assessment, 35 minutes.  We have added this information to the Results section. (Line 317)

There is completely missing contribution of this method for clinical use for cancer survivors. The aim of this new method – in future – find the new way how to make proper exercise under the supervision of a rehabilitation worker to improve physical coordination of patients, or how to reduce weight, other?   Response:  Cancers is a journal for which the readership is comprised not only of clinicians and clinician-scientists, but also basic and translational investigators. This manuscript was submitted for consideration in a special issue on the “Role of Lifestyle-related Factors in Cancer Survivorship,” for the purpose of disseminating our findings to cancer researchers who conduct anthropometric and physical function assessments in order to evaluate these outcomes within the context of clinical trials of lifestyle interventions.  While our hope is that oncologists find components of this study translatable to their practices (e.g., weight/waist measurements collected during telemedicine visits), our findings are perhaps more applicable in a research environment.  This approach is clarified in the Discussion section of the manuscript. (Line 443).

The description of the measurement method lacked an information how long the virtual and personal consultation itself takes. For the readers - clinical oncologists, the description of the method seems to be very long. Response: Thank you for the opportunity to respond.  The average time to complete the two virtual assessments was 25 minutes and the in-person assessment, 35 minutes.  We have added this information to the Results section. (Line 317)

There is completely missing contribution of this method for clinical use for cancer survivors. The aim of this new method – in future – find the new way how to make proper exercise under the supervision of a rehabilitation worker to improve physical coordination of patients, or how to reduce weight, other?   Response:  Cancers is a journal for which the readership is comprised not only of clinicians and clinician-scientists, but also basic and translational investigators. This manuscript was submitted for consideration in a special issue on the “Role of Lifestyle-related Factors in Cancer Survivorship,” for the purpose of disseminating our findings to cancer researchers who conduct anthropometric and physical function assessments in order to evaluate these outcomes within the context of clinical trials of lifestyle interventions.  While our hope is that oncologists find components of this study translatable to their practices (e.g., weight/waist measurements collected during telemedicine visits), our findings are perhaps more applicable in a research environment.  This approach is clarified in the Discussion section of the manuscript. (Line 443).

The description of the measurement method lacked an information how long the virtual and personal consultation itself takes. For the readers - clinical oncologists, the description of the method seems to be very long. Response: Thank you for the opportunity to respond.  The average time to complete the two virtual assessments was 25 minutes and the in-person assessment, 35 minutes.  We have added this information to the Results section. (Line 317)

There is completely missing contribution of this method for clinical use for cancer survivors. The aim of this new method – in future – find the new way how to make proper exercise under the supervision of a rehabilitation worker to improve physical coordination of patients, or how to reduce weight, other?   Response:  Cancers is a journal for which the readership is comprised not only of clinicians and clinician-scientists, but also basic and translational investigators. This manuscript was submitted for consideration in a special issue on the “Role of Lifestyle-related Factors in Cancer Survivorship,” for the purpose of disseminating our findings to cancer researchers who conduct anthropometric and physical function assessments in order to evaluate these outcomes within the context of clinical trials of lifestyle interventions.  While our hope is that oncologists find components of this study translatable to their practices (e.g., weight/waist measurements collected during telemedicine visits), our findings are perhaps more applicable in a research environment.  This approach is clarified in the Discussion section of the manuscript. (Line 443).

The description of the measurement method lacked an information how long the virtual and personal consultation itself takes. For the readers - clinical oncologists, the description of the method seems to be very long. Response: Thank you for the opportunity to respond.  The average time to complete the two virtual assessments was 25 minutes and the in-person assessment, 35 minutes.  We have added this information to the Results section. (Line 317)

There is completely missing contribution of this method for clinical use for cancer survivors. The aim of this new method – in future – find the new way how to make proper exercise under the supervision of a rehabilitation worker to improve physical coordination of patients, or how to reduce weight, other?   Response:  Cancers is a journal for which the readership is comprised not only of clinicians and clinician-scientists, but also basic and translational investigators. This manuscript was submitted for consideration in a special issue on the “Role of Lifestyle-related Factors in Cancer Survivorship,” for the purpose of disseminating our findings to cancer researchers who conduct anthropometric and physical function assessments in order to evaluate these outcomes within the context of clinical trials of lifestyle interventions.  While our hope is that oncologists find components of this study translatable to their practices (e.g., weight/waist measurements collected during telemedicine visits), our findings are perhaps more applicable in a research environment.  This approach is clarified in the Discussion section of the manuscript. (Line 443).

The description of the measurement method lacked an information how long the virtual and personal consultation itself takes. For the readers - clinical oncologists, the description of the method seems to be very long. Response: Thank you for the opportunity to respond.  The average time to complete the two virtual assessments was 25 minutes and the in-person assessment, 35 minutes.  We have added this information to the Results section. (Line 317)

There is completely missing contribution of this method for clinical use for cancer survivors. The aim of this new method – in future – find the new way how to make proper exercise under the supervision of a rehabilitation worker to improve physical coordination of patients, or how to reduce weight, other?   Response:  Cancers is a journal for which the readership is comprised not only of clinicians and clinician-scientists, but also basic and translational investigators. This manuscript was submitted for consideration in a special issue on the “Role of Lifestyle-related Factors in Cancer Survivorship,” for the purpose of disseminating our findings to cancer researchers who conduct anthropometric and physical function assessments in order to evaluate these outcomes within the context of clinical trials of lifestyle interventions.  While our hope is that oncologists find components of this study translatable to their practices (e.g., weight/waist measurements collected during telemedicine visits), our findings are perhaps more applicable in a research environment.  This approach is clarified in the Discussion section of the manuscript. (Line 443).

The description of the measurement method lacked an information how long the virtual and personal consultation itself takes. For the readers - clinical oncologists, the description of the method seems to be very long. Response: Thank you for the opportunity to respond.  The average time to complete the two virtual assessments was 25 minutes and the in-person assessment, 35 minutes.  We have added this information to the Results section. (Line 317)

There is completely missing contribution of this method for clinical use for cancer survivors. The aim of this new method – in future – find the new way how to make proper exercise under the supervision of a rehabilitation worker to improve physical coordination of patients, or how to reduce weight, other?   Response:  Cancers is a journal for which the readership is comprised not only of clinicians and clinician-scientists, but also basic and translational investigators. This manuscript was submitted for consideration in a special issue on the “Role of Lifestyle-related Factors in Cancer Survivorship,” for the purpose of disseminating our findings to cancer researchers who conduct anthropometric and physical function assessments in order to evaluate these outcomes within the context of clinical trials of lifestyle interventions.  While our hope is that oncologists find components of this study translatable to their practices (e.g., weight/waist measurements collected during telemedicine visits), our findings are perhaps more applicable in a research environment.  This approach is clarified in the Discussion section of the manuscript. (Line 443).

The description of the measurement method lacked an information how long the virtual and personal consultation itself takes. For the readers - clinical oncologists, the description of the method seems to be very long. Response: Thank you for the opportunity to respond.  The average time to complete the two virtual assessments was 25 minutes and the in-person assessment, 35 minutes.  We have added this information to the Results section. (Line 317)

There is completely missing contribution of this method for clinical use for cancer survivors. The aim of this new method – in future – find the new way how to make proper exercise under the supervision of a rehabilitation worker to improve physical coordination of patients, or how to reduce weight, other?   Response:  Cancers is a journal for which the readership is comprised not only of clinicians and clinician-scientists, but also basic and translational investigators. This manuscript was submitted for consideration in a special issue on the “Role of Lifestyle-related Factors in Cancer Survivorship,” for the purpose of disseminating our findings to cancer researchers who conduct anthropometric and physical function assessments in order to evaluate these outcomes within the context of clinical trials of lifestyle interventions.  While our hope is that oncologists find components of this study translatable to their practices (e.g., weight/waist measurements collected during telemedicine visits), our findings are perhaps more applicable in a research environment.  This approach is clarified in the Discussion section of the manuscript. (Line 443).

The description of the measurement method lacked an information how long the virtual and personal consultation itself takes. For the readers - clinical oncologists, the description of the method seems to be very long. Response: Thank you for the opportunity to respond.  The average time to complete the two virtual assessments was 25 minutes and the in-person assessment, 35 minutes.  We have added this information to the Results section. (Line 317)

There is completely missing contribution of this method for clinical use for cancer survivors. The aim of this new method – in future – find the new way how to make proper exercise under the supervision of a rehabilitation worker to improve physical coordination of patients, or how to reduce weight, other?   Response:  Cancers is a journal for which the readership is comprised not only of clinicians and clinician-scientists, but also basic and translational investigators. This manuscript was submitted for consideration in a special issue on the “Role of Lifestyle-related Factors in Cancer Survivorship,” for the purpose of disseminating our findings to cancer researchers who conduct anthropometric and physical function assessments in order to evaluate these outcomes within the context of clinical trials of lifestyle interventions.  While our hope is that oncologists find components of this study translatable to their practices (e.g., weight/waist measurements collected during telemedicine visits), our findings are perhaps more applicable in a research environment.  This approach is clarified in the Discussion section of the manuscript. (Line 443).

The description of the measurement method lacked an information how long the virtual and personal consultation itself takes. For the readers - clinical oncologists, the description of the method seems to be very long. Response: Thank you for the opportunity to respond.  The average time to complete the two virtual assessments was 25 minutes and the in-person assessment, 35 minutes.  We have added this information to the Results section. (Line 317)

There is completely missing contribution of this method for clinical use for cancer survivors. The aim of this new method – in future – find the new way how to make proper exercise under the supervision of a rehabilitation worker to improve physical coordination of patients, or how to reduce weight, other?   Response:  Cancers is a journal for which the readership is comprised not only of clinicians and clinician-scientists, but also basic and translational investigators. This manuscript was submitted for consideration in a special issue on the “Role of Lifestyle-related Factors in Cancer Survivorship,” for the purpose of disseminating our findings to cancer researchers who conduct anthropometric and physical function assessments in order to evaluate these outcomes within the context of clinical trials of lifestyle interventions.  While our hope is that oncologists find components of this study translatable to their practices (e.g., weight/waist measurements collected during telemedicine visits), our findings are perhaps more applicable in a research environment.  This approach is clarified in the Discussion section of the manuscript. (Line 443).

The description of the measurement method lacked an information how long the virtual and personal consultation itself takes. For the readers - clinical oncologists, the description of the method seems to be very long. Response: Thank you for the opportunity to respond.  The average time to complete the two virtual assessments was 25 minutes and the in-person assessment, 35 minutes.  We have added this information to the Results section. (Line 317)

There is completely missing contribution of this method for clinical use for cancer survivors. The aim of this new method – in future – find the new way how to make proper exercise under the supervision of a rehabilitation worker to improve physical coordination of patients, or how to reduce weight, other?   Response:  Cancers is a journal for which the readership is comprised not only of clinicians and clinician-scientists, but also basic and translational investigators. This manuscript was submitted for consideration in a special issue on the “Role of Lifestyle-related Factors in Cancer Survivorship,” for the purpose of disseminating our findings to cancer researchers who conduct anthropometric and physical function assessments in order to evaluate these outcomes within the context of clinical trials of lifestyle interventions.  While our hope is that oncologists find components of this study translatable to their practices (e.g., weight/waist measurements collected during telemedicine visits), our findings are perhaps more applicable in a research environment.  This approach is clarified in the Discussion section of the manuscript. (Line 443).